# Obesity Is Associated with Increased F₂-Isoprostanes and IL-6 in Black Women

**Mohammad Saleem** , **Paul D. Kastner, Pouya Mehr, Ginger L. Milne** , **Jeanne A. Ishimwe, Jennifer H. Park, Cyndya A. Shibao * and Annet Kirabo ***

Department of Medicine, Division of Clinical Pharmacology, Room 536 Robinson Research Building, Vanderbilt University Medical Center, Nashville, TN 37212-8802, USA
* Correspondence: cyndya.shibao@vumc.edu (C.A.S.); annet.Kirabo@vumc.org (A.K.)

**Abstract:** Obesity affects over 40% of the adult population and is a major risk factor for morbidity and mortality due to cardiovascular disease. Black women have one of the highest prevalences of obesity, insulin resistance, hypertension, and cardiovascular events in the US. We previously found that free radical-mediated lipid peroxidation contributes to IL-6 production in dendritic cells leading to inflammation and hypertension. Thus, we hypothesized that F₂-isoprostanes (F₂-IsoPs), products and biomarkers of endogenous lipid peroxidation, contribute to increased inflammation and IL-6 production among obese Black women. We studied a total of 88 obese Black women of age $42.0 \pm 9.8$ years, weight $102 \pm 16$ kg, and body mass index (BMI) $37.68 \pm 5.08$. Systolic and diastolic blood pressure were $124 \pm 14/76.2 \pm 9.9$ mmHg, heart rate was $68.31 \pm 10.26$ beats/min, and fasting insulin was $15.0 \pm 8.7$ uU/mL. Plasma F₂-IsoPs were measured using gas chromatography/negative ion chemical ionization mass spectrometry (GC/NICI-MS). Plasma cytokines, including IL-6, IL-8, IL-10, IL-1β, TNF-a, and C-reactive proteins were measured using multiplex Luminex technology. Anthropometric measurements were performed using dual-energy X-ray absorptiometry. Using Pearson's correlation analysis, we found that BMI was positively correlated with plasma F2-IsoPs, while inversely correlated with insulin sensitivity in obese Black women. Further, F₂-IsoPs were positively correlated with inflammatory marker IL-6 levels while negatively correlated with anti-inflammatory marker IL-10. In addition, we found that plasma F₂-IsoPs levels were significantly associated with reduced insulin sensitivity. These results suggest that F₂-IsoPs may be associated with obesity-induced cardiovascular risk in Black women by increasing the production of inflammatory cytokine IL-6 and decreasing the production of anti-inflammatory IL-10.

**Keywords:** obesity; black women; inflammation; lipid oxidation; oxidative stress; F2-isoprostanes

## 1. Introduction

Obesity is now considered an epidemic worldwide [1] and is a strong risk factor for inflammation, insulin resistance, type 2 diabetes mellitus (T2DM), and cardiovascular diseases (CVD) [2–5]. The disproportionately high incidence of obesity in Blacks was recognized even before the emergence of obesity as a critical health issue in the US population [6–9]; however, the concurrent strategies to reduce obesity levels in the minority population were not developed [10]. The US National Health and Nutritional Examination Survey (NHANES, 2011–2012) analyses reported that 76.2% of Black adults ages 20 years or older had (body mass index) BMI levels of 25 or greater compared to 67.2% of Whites. At the BMI threshold of 30, the prevalence of obesity was 47.8% in Blacks and 32.6% in Whites. The prevalence of obesity at the higher cut-off of BMI > 35 was more than double (i.e., grade 2 and grade 3 obesity combined), which was 23.3% in Blacks and 11.2% in Whites [11].

According to National Health Statistics Report June 2021, non-Hispanic Black women had the highest prevalence of obesity compared with women of other races and Hispanic-origin groups. Black women are disproportionately impacted by obesity [12] and related

comorbidities such as diabetes [13] and hypertension [14]. With standard weight loss interventions, Black women tend to lose less and slower weight than Black men [15,16] and White women [15–18]. Therefore, developing effective prevention strategies and weight maintenance remains an important research priority [19].

Obesity is considered a low-grade chronic inflammatory condition, and several lines of evidence show the involvement of inflammatory cytokines, including tumor necrosis factor-alpha (TNF-$\alpha$), interleukin-6 (IL-6), and interleukin-8 (IL-8) [20,21], and anti-inflammatory cytokines IL-10 [22,23] in obesity. Moreover, previous studies have found that obesity contributes to diabetes [1]. However, the contribution of the specific cytokine to inflammation and diabetes in obese Black women is not well known. In previous studies, we found that IsoLGs, which are formed in the F2-IsoPs pathway of lipid peroxidation, accumulate in antigen-presenting cells in murine models of obesity [24], leading to their activation and increased production of IL-6 [25]. We also found that elevated plasma levels of F2-IsoPs are associated with increased inflammation and hypertension [25]. Therefore, we wanted to investigate how cytokines and oxidative stress are associated with inflammation and prediabetic indices in obese Black women.

Oxidative stress, defined as an imbalance between the production and elimination of reactive oxygen species (ROS), is also associated with various inflammation-related human diseases including obesity, diabetes, hypertension, and aging [26–29]. ROS can oxidize polyunsaturated fatty acids that subsequently synthesize oxidation products including F2-isoprostanes (F2-IsoPs) (Figure 1B). F2-IsoPs are one of the most well-studied and reliable biomarkers for assessing oxidative stress in human studies [30]. Plasma and urinary F2-IsoPs have been used to evaluate the effect of chronic oxidative stress on the endothelium and other tissues. A recent study showed increased levels of F2-IsoPs and IL-6 in obese women [31], while several other studies showed a correlation between IL-6 and F2-IsoPs correlation between inflammatory marker IL-6 and oxidative stress markers F2-IsoPs [32]. Several studies have demonstrated strong relationships between biomarkers of oxidative stress and elevated inflammatory cytokines during several disease conditions. This association has been found among smokers, patients with acute kidney injury (AKI), and obstructive sleep apnea [33–35].

Much of the literature regarding the treatment of obesity revolve around the use of models and data derived from Caucasians. However, the disproportionate impact of obesity on Black women, in particular, makes it imperative to develop differential strategies for effective treatment practices for Blacks. Thus, herein we specifically investigated the relationship between biomarkers for obesity, chronic inflammation, and oxidative stress in obese Black women.

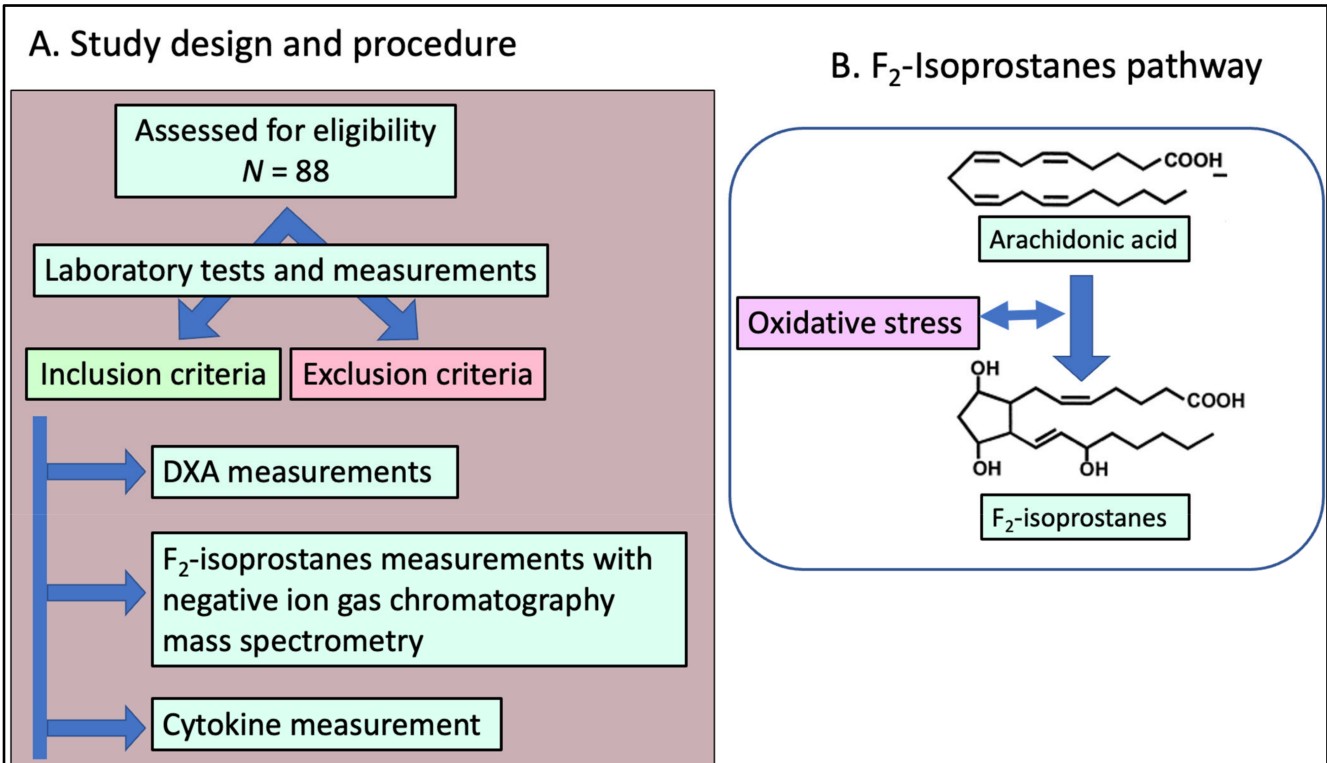

**Figure 1.** Study design and $F_2$-IsoPs pathway. (**A**) Workflow of the study. This was an observational study. A total of 88 Black women were enrolled in the study. Laboratory tests and anthropometric measurements were performed to include or exclude the participants. (**B**) Depiction of $F_2$-isoprostane synthesis pathway.

## 2. Materials and Methods

### 2.1. Study Design and Population Enrollment

A total of 88 obese Black women volunteers were enrolled in this study. The baseline characteristics of these patients are presented in Table 1. The participants were recruited from referrals to the Vanderbilt Autonomic Dysfunction Center. The study was approved by an institutional review board (Vanderbilt Human Research Protection Program), and all participants gave written informed consent. Further, the study was conducted following institutional guidelines and adhered to the principles of the Declaration of Helsinki and Title 45 of the US Code of Federal Regulations (Part 46, Protection of Human Subjects). The studies were registered on ClinicalTrials.gov, identifier NCT02365285.

The eligibility criteria included obese women (as defined by BMI) between 30 and 45 kg/m$^2$), aged 18 to 60 years old of the Black race. The race was self-defined, but only subjects who reported both parents of the same race were included. The pregnant or breast-feeding women, individuals diagnosed with type 2 diabetes mellitus, hypertension or any cardiovascular disease, impaired renal function (glomerular filtration rate, GFR < 60%), impaired hepatic function (abnormal liver function test), or had a history of alcohol or drug abuse were excluded. We also excluded individuals who used potent inhibitors of cytochrome P450 (CYP3A4), cytochrome P450 (CYP2D6), AchE inhibitors such as pyridostigmine, bethanechol, or had a significant weight change ≥5% in the previous three months. We outlined the workflow of the study in Figure 1A.

Subjects who fulfilled the eligibility criteria were admitted to the clinical research center (CRC) for the study. Before admission, subjects were asked to collect 24 h urine for sodium, creatinine, and $F_2$-IsoPs measurements. An intravenous catheter was placed in one arm for blood sampling. The demographic characteristics are presented in Table 1.

**Table 1.** Demographic characteristics of subjects.

| Variable | N | |
|---|---|---|
| Age | 87 | $42.61 \pm 10.02$ |
| Height (cm) | 83 | $162.9 \pm 5.2$ |
| Weight (kg) | 83 | $102 \pm 15$ |
| BMI, kg/m$^2$ | 83 | $38.0 \pm 5.0$ |
| Baseline SBP, mmHg | 77 | $124 \pm 14$ |
| Baseline DBP, mmHg | 77 | $76.2 \pm 9.9$ |
| Heart rate, bpm | 80 | $68.31 \pm 10.26$ |
| Sodium | 87 | $139.0 \pm 2.4$ |
| Potassium | 87 | $3.93 \pm 0.36$ |
| Insulin, mU/ml | 50 | $15.0 \pm 8.7$ |
| Fasting insulin (microU/L) | 33 | $8.4 \pm 7.3$ |
| Fasting glucose (nmol/L) | 42 | $97 \pm 10$ |
| Fat-free mass, kg | 56 | $52.8 \pm 6$ |
| Percent fat mass | 56 | $47.1 \pm 4.5$ |
| HDL, mg/dL | 72 | $46.5 \pm 10.9$ |
| LDL mg/dL | 72 | $110.3 \pm 32.9$ |
| Triglycerides, mg/dL | 81 | $89.3 \pm 49.2$ |
| Total cholesterol, mg/dL | 72 | $175.4 \pm 35.8$ |
| Waist size, cm | 81 | $109.2 \pm 11.5$ |
| Creatinine, mg/dL | 88 | $0.8 \pm 0.1$ |
| BUN, mg/dL | 87 | $0.81 \pm 0.11$ |
| Isoprostanes, ng/mL | 88 | $0.050 \pm 0.021$ |

BMI = body mass index; SBP = systolic blood pressure; DBP = diastolic blood pressure; HDL = high-density lipoprotein; LDL = low-density lipoprotein; BUN = blood urea nitrogen. The third column shows the average and standard deviation for each variable.

*2.2. Blood Sample Collection*

The blood samples were collected in prelabelled chilled ethylenediaminetetraacetic acid (EDTA) tubes and were immediately centrifuged to separate the plasma and stored at $-80$ °C. We measured plasma glucose at the bedside with a glucose analyzer (YSI Life Sciences, Yellow Springs, OH, USA). Plasma insulin concentrations were determined by radioimmunoassay (Millipore, St. Charles, MO, USA). Insulin sensitivity, beta-cell response, and disposition index measurements were calculated using a frequently-sampled intravenous glucose tolerance test described by Bergman RN et al [36].

*2.3. Measurement of $F_2$-IsoPs in Plasma and Peripheral Blood Mononuclear Cells (PBMCs)*

The levels of $F_2$-IsoPs were measured in plasma using negative ion gas chromatography-mass spectrometry by the Vanderbilt University Eicosanoid Core Laboratory as previously described [37,38]. Similarly, levels of $F_2$-IsoPs in PBMCs were measured as previously described [39].

## 2.4. Dual-Energy X-ray Absorptiometry

Total and regional body composition was acquired by a certified densitometric using a Lunar iDXA whole-body scanner (GE Healthcare, Madison, WI, USA) with the enCore 2007 software (version 11.4). Before each acquisition, the scanner was phantom-calibrated according to the manufacturer's instructions [40]. Scans were imported into an updated version of the software (version 13.6) and reanalyzed using the CoreScan algorithm, which provides automatic segmentation of estimated visceral adipose tissue (e-VAT) from total abdominal fat within the android region. E-VAT mass (g) was automatically transposed into volume ($cm^3$) using a constant correction factor (0.94 g/mL) that is consistent with the density of the adipose tissue [41].

## 2.5. Resting Energy Expenditure (REE)

REE was assessed as previously described [42]. Briefly, subjects were studied after 12-h fasting, and only clear fluids were allowed after 8:00 p.m. the night before. Intense physical activity was not permitted the day before. Women were studied in the follicular phase of their menstrual cycle (days 1 to 12). The subject rested quietly supine at an ambient temperature of 21 °C for 30 min before testing. An open-circuit indirect calorimeter assessed REE with a ventilated canopy or a face tent device (CPX/D system, Medical Graphics Corporation). Only the last 20 min of a 40 min measurement period were analyzed. The respiratory quotient was used for quality control. Basal REE was measured between 7:00 and 8:00 a.m. and between 11:00 and 12:00 a.m. This sequential design was selected to use each subject as their own control. A single operator performed all the studies.

## 2.6. Cytokine Measurement

Levels of plasma cytokines (IL-6, IL-8, IL-10, TNF-alpha, and C-reactive proteins) were measured with multiplex Luminex technology using an x-map MagPix system (Millipex map human cytokine/chemokine magnetic bead panel, Millipore Sigma, MS, USA).

## 2.7. Statistics

We employed standard graphing and screening techniques to detect outliers and to ensure data accuracy. The data were assessed for normality. We analyzed the data using R version 3.5.3 software and expressed as the mean $\pm$ SD throughout the manuscript unless otherwise indicated. Pearson's rank correlation was used to determine the correlation using GraphPad prism 9.4.1. $p < 0.05$ with two-tailed analysis was considered statistically significant.

## 3. Results

### 3.1. Obesity Is Associated with Increased Pro-Inflammatory Cytokine IL-6 in Black Women

To determine if the inflammatory cytokines associated with obesity in Black women, we performed Pearson's correlation analysis between BMI and various cytokines, including IL-6, IL-8, IL-10, IL-1b, TNF-a, and C-reactive protein. The analyses showed high BMI correlated with higher plasma IL-6 (Figure 2A R = 0.14, $p = 0.04$). However, we did not find any correlation of BMI with IL-8, IL-10, IL-1b, TNF-a, or C-reactive protein (Figure 2B–F).

### 3.2. The $F_2$-IsoPs Are Associated with BMI, Lipoproteins, and Hypertension in Black Women

The data analyses showed a positive association between BMI and increased $F_2$-IsoPs levels (Figure 3A, r = 0.23, $p = 0.04$). In addition, we found a positive correlation between $F_2$-IsoPs and percent fat (Figure 3B, r = 0.30, $p = 0.03$), arm percent fat (Supplementary Figure S4A, r = 0.34, $p = 0.02$), leg percent fat (Supplementary Figure S4B, r = 0.37, $p = 0.01$), total leg mass (Supplementary Figure S4C, r = 0.29, $p = 0.05$), and thigh circumference (Supplementary Figure S4D, r = 0.26, $p = 0.03$). To determine if $F_2$-IsoPs were associated with cardiovascular risk biomarkers, we performed Pearson's correlation analyses between $F_2$-IsoPs, high-density lipoproteins (HDL), and low-density lipoproteins (LDL) in patients with BMI between 30 and 40. There was an inverse correlation between $F_2$-IsoPs and HDL (Figure 3C, r = $-33$, $p = 0.04$), while there was a positive correlation

trend between $F_2$-IsoPs and LDL (Figure 3D, r = −18, $p = 0.14$). To see if the $F_2$-IsoPs levels in PBMCs change with blood pressure changes, $F_2$-IsoPs were measured in PBMCs in a separate cohort of obese women. We dichotomize systolic blood pressure below 120 and 120 or above. Our results show that women with SBP 120 mmHg or above exhibited higher levels of $F_2$-IsoPs in PBMCs (Figure 3E, $p = 0.05$). These findings indicate that $F_2$-IsoPs may mainly play a role in cardiometabolic disease associated with obesity in Black women.

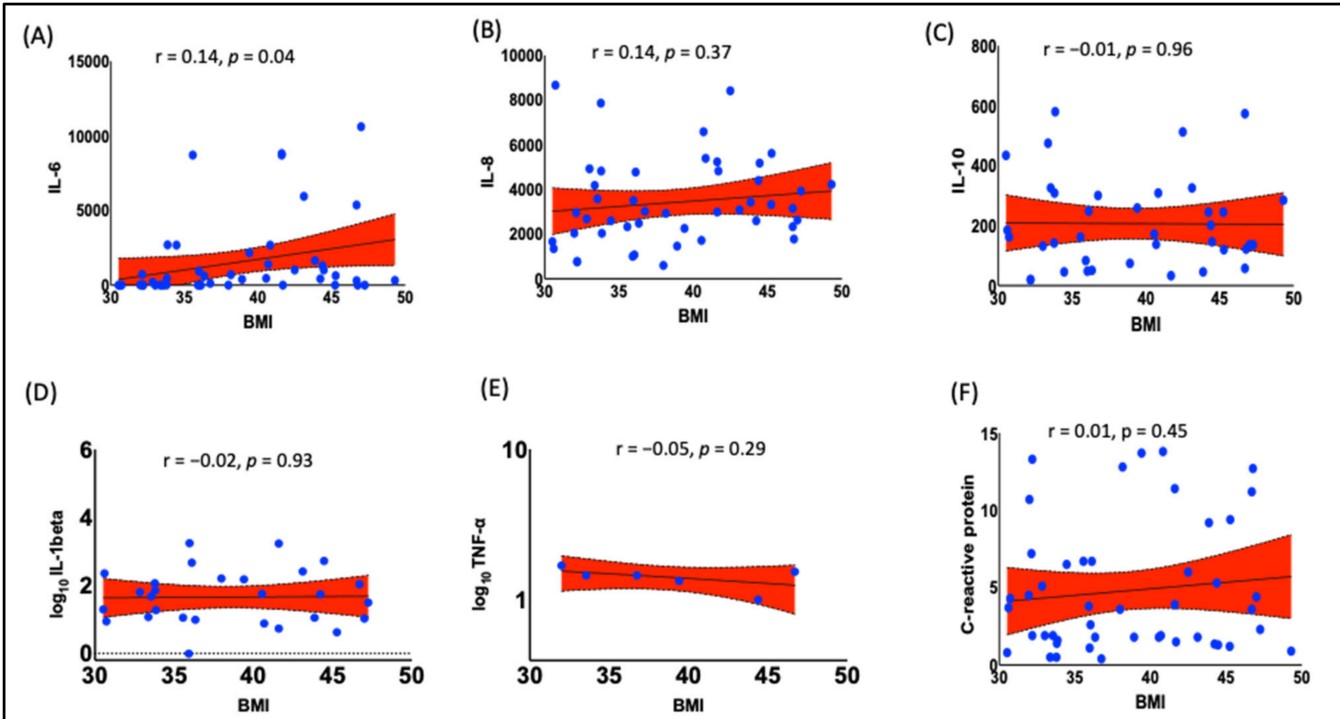

**Figure 2.** Relationship between BMI and inflammatory markers. There was a positive correlation between (**A**) BMI and IL-6 (r = 0.14, $p = 0.04$); however, there was no correlation between BMI and other inflammatory cytokines (**B**) IL-8, (**C**) IL-10, (**D**) IL-1beta, (**E**) TNF-alpha, and (**F**) C-reactive protein. Pearson's correlation analysis was employed to determine the correlation. $p < 0.05$ was considered a statistically significant correlation. IL, interleukin; BMI, body mass index; TNF, tumor necrosis factor.

### 3.3. Plasma $F_2$-IsoPs Are Associated with IL-6 and IL-10 in Obese Black Women

Our data analyses showed that IL-6 was positively correlated with increased $F_2$-IsoPs (Figure 4A r = 0.46, $p = 0.003$), while negatively correlated with anti-inflammatory cytokine IL-10 (Figure 4B, R = −0.38, $p = 0.02$). Like BMI, levels of $F_2$-IsoPs were not correlated with other cytokines such as IL-8, IL-1beta, TNF-alpha, and C-reactive protein (Figure 4C–F).

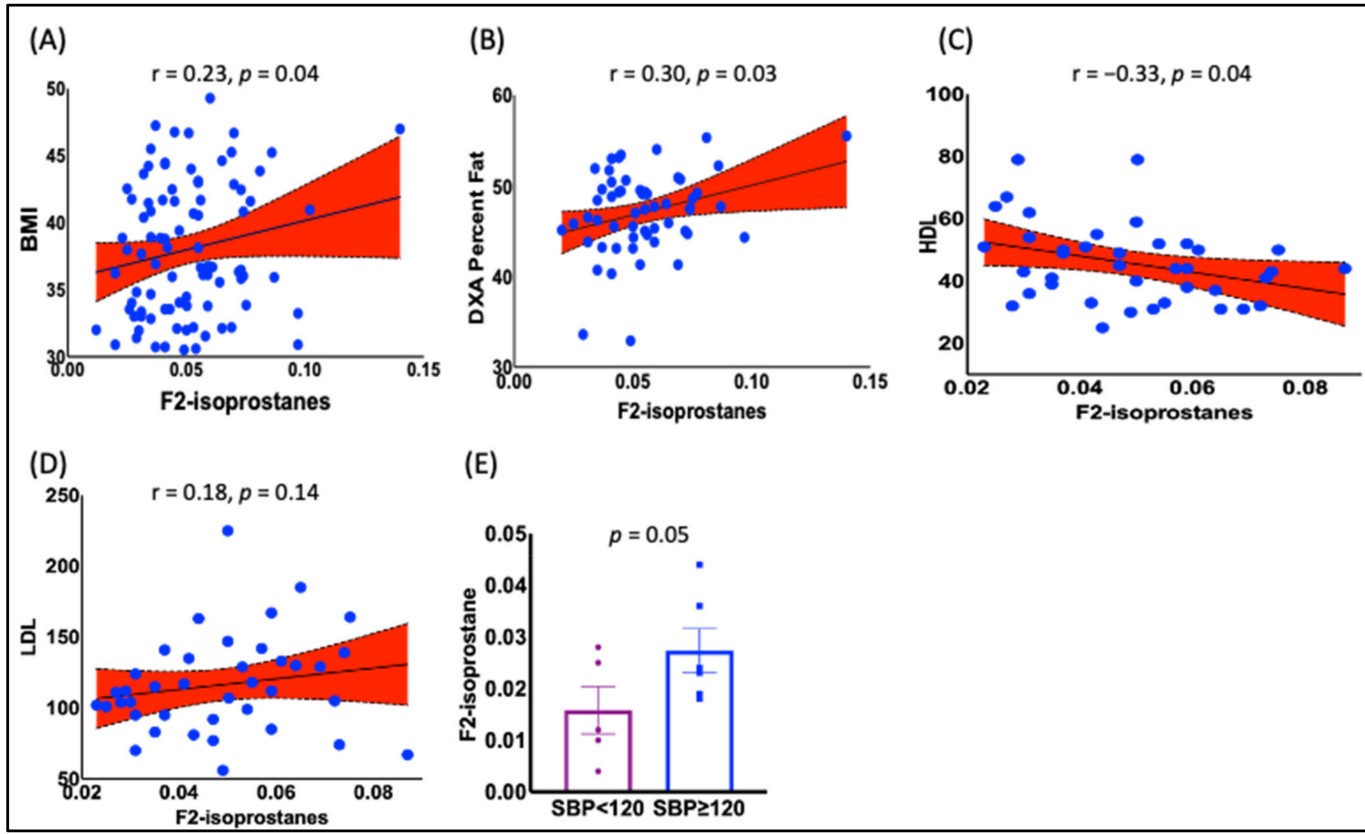

**Figure 3.** Relationship between BMI and other anthropometric DXA measurements and $F_2$-IsoPs. Our data show a positive correlation between (**A**) BMI and $F_2$-IsoPs (r = 0.23, *p* = 0.04). Anthropometric parameters, such as (**B**) total percent fat (r = 0.30, *p* = 0.03), were also correlated with $F_2$-isoprostanes. (**C**) HDL levels were inversely correlated (r = −0.33, *p* = 0.04), while (**D**) LDL showed a positive trend (r = 0.18, *p* = 0.14). (**E**) Hypertensive people showed increased $F_2$-IsoPs levels in PBMCs compared to normotensive people (*p* = 0.05). Pearson's correlation analysis was employed to determine the correlation. *p* < 0.05 was considered a statistically significant correlation. A *t*-test was employed to compare normotensives and hypertensives. BMI, body mass index; HDL, high-density lipoproteins; LDL, low-density lipoprotein; SBP, systolic blood pressure.

### 3.4. Prediabetic Indices Are Correlated with BMI

The data analyses found a significant inverse correlation between higher BMI and insulin sensitivity (Figure 5A, r = 0.38., *p* = 0.03), as well as disposition index (Figure 5B, r = 0.51, *p* = 0.003). Further, estimated visceral adipose tissue [(e-VAT, g, and cm$^3$) (Figure 5C, r = 0.38, *p* = 0.03; Figure 5D, r = 0.38, *p* = 0.03)] and fat mass were negatively correlated with insulin sensitivity (Figure 5E, r = −0.41, *p* = 0.02). To determine if $F_2$-IsoPs are associated with reduced insulin sensitivity, we performed additional analyses in which we found an inverse correlation between $F_2$-IsoPs and insulin sensitivity (Figure 5F, r = −0.41., *p* = 0.02).

*3.5. Obesity Is Correlated with Resting Energy Expenditure (REE) and Fractional Concentrations of Expired $O_2$ and $CO_2$*

Pearson correlation analysis showed that BMI positively correlated with $VO_{2(STPD)}$ (Figure 6A, A = 0.60, *p* = 0.001), $VCO_{2(STPD)}$, (Figure 6B, R = 0.56, *p* = 0.003), $V_E$ (Figure 6C, r = 0.47, *p* = 0.02), and REE (Figure 6D, r = 0.60, *p* = 0.001), while inversely correlated with metabolic equivalents [METs, (Figure 6E, r = −0.45, *p* = 0.02)], and $VO_2$, (Figure 6F, r = 0.45, *p* = 0.02). Most importantly, $F_2$-IsoPs was positively correlated with $F_ECO_2$ [(fractional concentrations of expired carbon dioxide (Figure 6G, r = 0.54, *p* = 004)], while inversely with $F_EO_2$ [fractional concentrations of expired oxygen, (Figure 6H, r = −0.44, *p* = 0.02)], suggesting that more $CO_2$ is being generated and expired due to higher cellular metabolism and less $O_2$ is being expired, and the remaining $O_2$ is being used to generate other free radical such as superoxide ($O_2{}^-$). $F_2$-IsoP did not show any correlation with REE (Figure 6I, r = 0.06, *p* = 0.76).

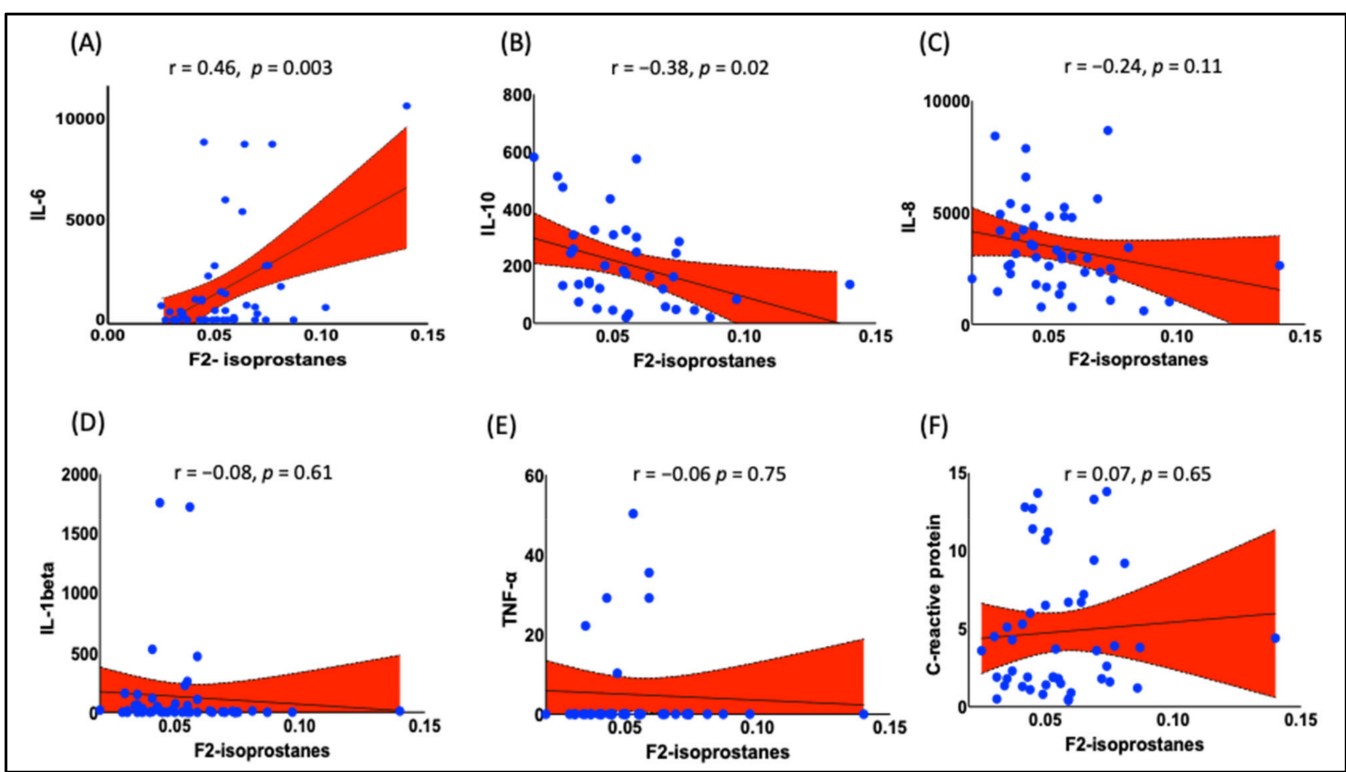

**Figure 4.** Relationship between $F_2$-IsoPs and inflammatory markers. Data analysis showed a positive correlation between $F_2$-IsoPs and inflammatory marker (**A**) IL-6 (r = 0.46, *p* = 0.003), while an inverse correlation with the anti-inflammatory marker (**B**) IL-10 (r = −0.38, *p* = 0.02). However, there was no correlation between $F_2$-IsoPs and other inflammatory cytokines (**C**) IL-8, (**D**) IL-1beta, (**E**) TNF-alpha, and (**F**) C-reactive protein. Pearson's correlation analysis was employed to determine the correlation. *p* < 0.05 was considered a statistically significant correlation. IL, interleukin; TNF, tumor necrosis factor.

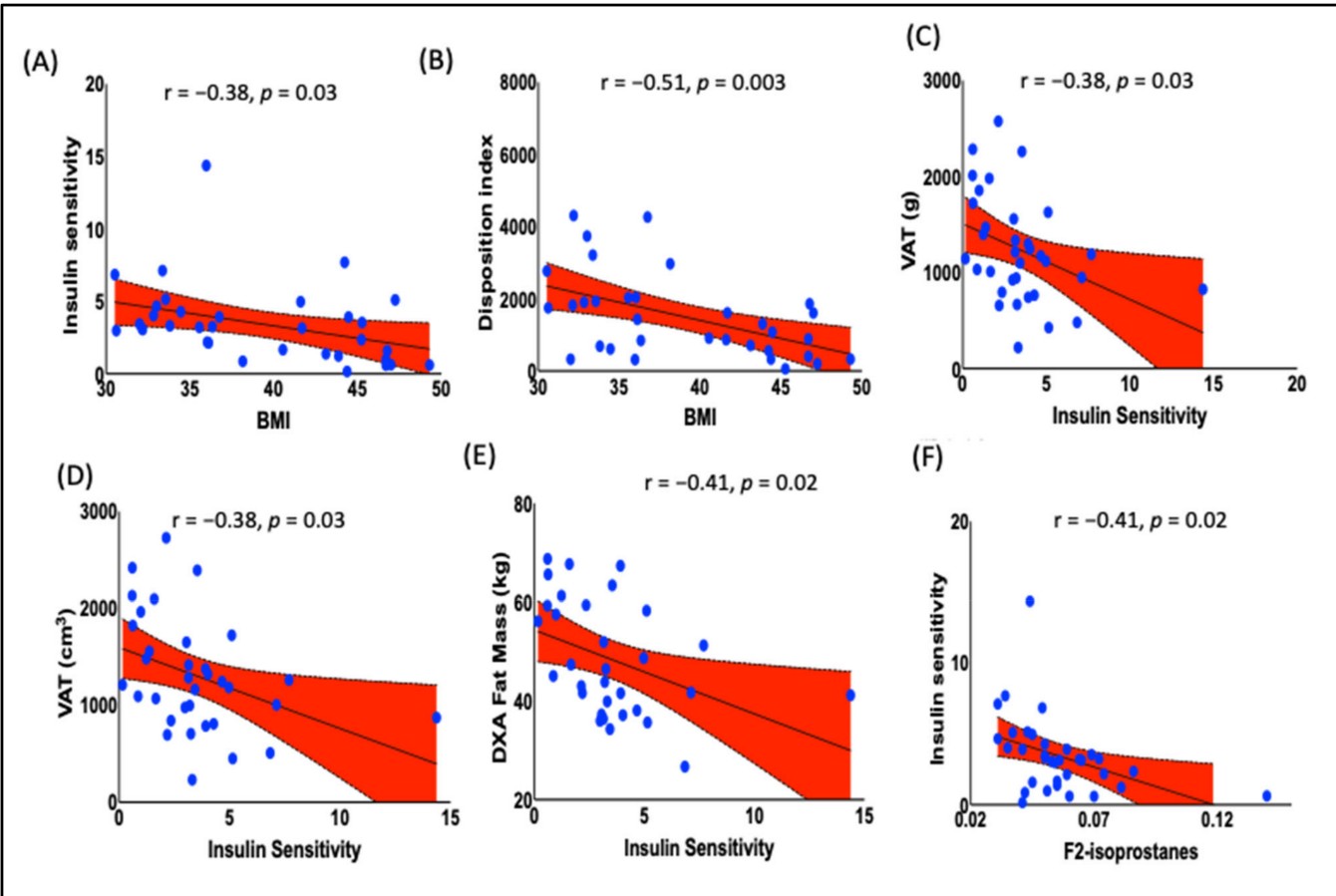

**Figure 5.** Relationship of prediabetic indices with BMI and other anthropometrics. BMI was inversely correlated with prediabetic indices (**A**) insulin sensitivity ($r = -0.3753$, $p = 0.0287$) and (**B**) disposition index ($r = -0.51$, $p = 0.003$). Moreover, (**C**) e-VAT (g, $r = -0.38$, $p = 0.03$), (**D**) e-VAT (cm$^3$, $r = -0.38$, $p = 0.03$), (**E**) DXA fat mass ($r = -0.41$, $p = 0.02$), and (**F**) F$_2$-IsoPs ($r = -0.41$, $p = 0.02$) levels were inversely correlated with insulin sensitivity. Pearson's correlation analysis was employed to determine the correlation. $p < 0.05$ was considered a statistically significant correlation. BMI, body mass index; e-VAT, estimated visceral adipose tissue.

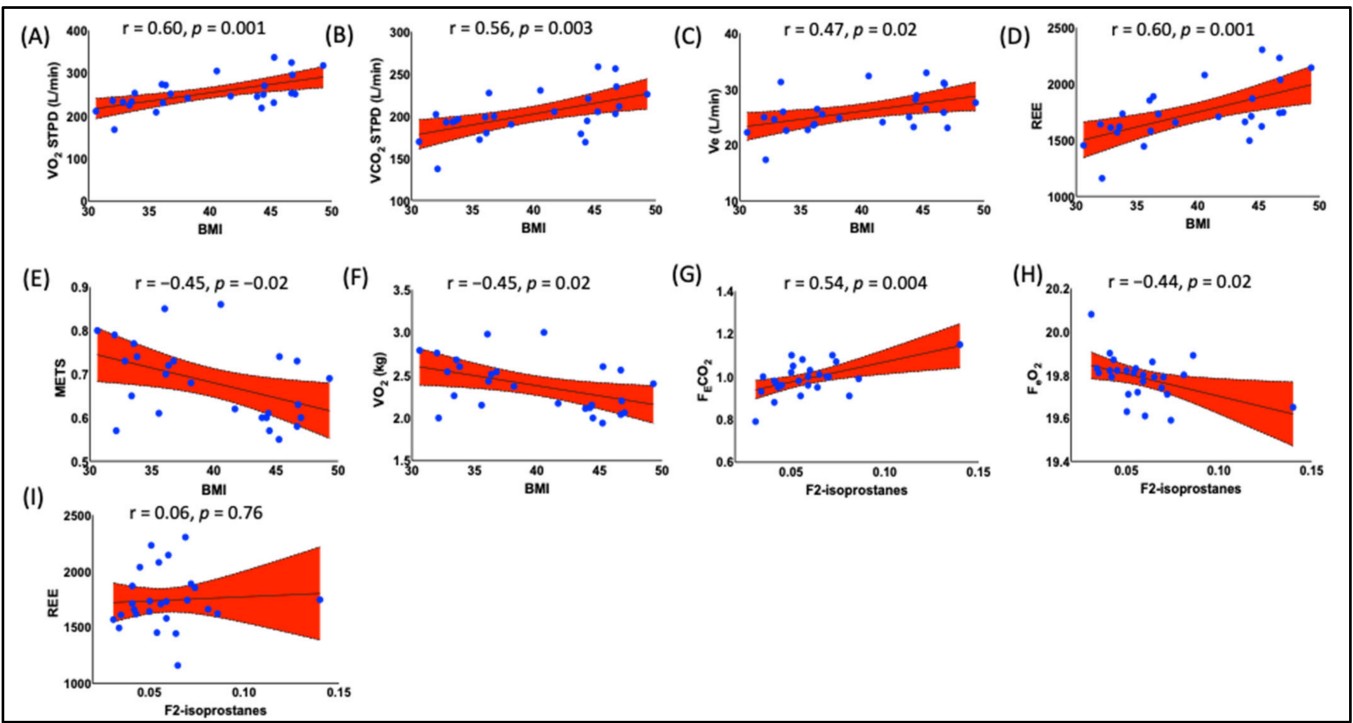

**Figure 6.** Relationship of specific body circumferences with BMI and F$_2$-IsoPs. We found a strong positive correlation between BMI and (**A**) VO$_{2(STPD)}$ (l/min, r = 0.60, $p$ = 0.001), (**B**) VCO$_{2(STPD)}$ (l/min, r = 0.56, $p$ = 0.003), (**C**) Ve (l/min, r = 0.47, $p$ = 0.02), and (**D**) REE (kcal, r = 0.60, $p$ = 0.0162), while the inverse correlation with (**E**) METs (r = −0.45, $p$ = 0.02), and (**F**) VO$_2$ (kg, r = −0.45, $p$ = 0.02). There was a positive correlation between F$_2$-IsoPs and (**G**) FeCO$_2$ (r = 0.54, $p$ = 004), while an inverse correlation with (**H**) FeO$_2$ (r = −0.44, $p$ = 0.02), and no correlation with (**I**) REE (r = 0.06, $p$ = 0.76). Pearson's correlation analysis was employed to determine the correlation. $p < 0.05$ was considered a statistically significant correlation. BMI, body mass index; VO$_2$, the volume of oxygen uptake; VCO$_2$, the volume of carbon dioxide expired; V$_E$, the volume of expired air; REE, resting energy expenditure; MET, metabolite equivalent; F$_E$CO$_2$, fractional concentrations of expired carbon dioxide; F$_E$O$_2$, fractional concentrations of expired oxygen.

## 4. Discussion

A recent report from the Centers for Disease Control and Prevention (CDC, 2020) stated that obesity disproportionately (https://www.cdc.gov/obesity/data/prevalence-maps.html, accessed on 25 May 2022) affects Blacks in the United States. However, the higher prevalence of obesity in Blacks is confined to women. Black women have a 55% prevalence of obesity, compared to 37% of Black men and 38% of both sexes of non-Hispanic Whites when data were analyzed by race and sex (https://www.cdc.gov/nchs/data/databriefs/db288.pdf, accessed on 25 May 2022). The etiology and pathophysiology of the disproportionate prevalence of obesity in Black women are still unclear. In this observational study, we found that obesity is associated with elevated levels of the oxidative stress biomarkers F$_2$-IsoPs and inflammatory cytokine IL-6. More importantly, we also found that F$_2$-IsoPs positively correlated with the levels of IL-6 while inversely correlated with anti-inflammatory cytokine IL-10, suggesting a causative role of oxidative stress in chronic inflammation in obese Black women. In addition to the main findings, our data show an inverse relationship between F$_2$-IsoPs and insulin sensitivity.

There are conflicting reports about insulin sensitivity in obese Black women [43–46]. Some reports show decreased [43] while others show elevated [44–46] insulin sensitivity in obese Black women. Our results agree with the former, and it is unclear why this discrepancy exists. F$_2$-IsoPs have been implicated in insulin resistance, T2D, and cardiometabolic disease [47–49], although some contrasting evidence exists [50]. Studies in obese animals

showed increased plasma $F_2$-IsoPs with hyperinsulinemia and hyperglycemia, which were attenuated after antioxidant and vitamin E supplement [51]. Similarly, insulin-resistant obese patients exhibited high levels of urinary $F_2$-IsoPs and oxidized low-density lipoprotein (LDL), when compared with normal-weight individuals [52]. Racial difference studies showed that Blacks exhibit lower $F_2$-IsoPs than Whites in urine and plasma [53–55]. In contrast to ours, however, these studies were conducted in both males and females. Fat accumulation is associated with the markers of systemic oxidative stress [56–58], and increased oxidative stress in accumulated fat is, to some degree, the underlying cause of dysregulation of adipocytokines and the development of metabolic syndrome [56]. Studies in diabetic patients showed that antioxidant supplementation reduced the levels of $F_2$-IsoPs [59,60]. These studies, together with our results, suggest that oxidative stress is associated with reduced insulin sensitivity, which is an initial step in the inception of T2D, in obese Black women.

Numerous epidemiological cross-sectional and observational studies demonstrated a positive correlation between $F_2$-IsoPs and obesity and increased risk of T2D [57,61,62]. However, conflicting reports show an inverse relationship, meaning that $F_2$-IsoPs levels are decreased with weight gain in Blacks and Whites [50,53,54]. The reason for this inverse relationship has been postulated to be due to a compensatory mechanism in obese patients and that negative energy balance (weight loss) leads to a decrease in energy expenditure, while positive energy balance (weight gain) leads to an increase in energy expenditure [63]. Since fat mass is the predominant component of body mass changes, fat oxidation plays an essential function in the physiological control of the energy balance [64–66]. Furthermore, more efficient fat oxidation in a non-obese individual alleviates the risk of weight gain and, thereby, the risks of obesity and T2D, thus the increased levels of urinary $F_2$-IsoPs [67,68]. However, these studies are very few and were conducted only in a specific group of people, Pima Indians, which might differ from our study subjects, obese Black women. In contrast to these studies but in consensus with most studies, our study indicated that increased plasma $F_2$-IsoPs and resting energy expenditure (REE) positively correlate with BMI in obese Black women (Figures 3A and 6D).

Several studies have demonstrated strong relationships between oxidative stress and elevated inflammatory markers during several disease conditions. A study in smokers exhibited elevated levels of cytokine IL-6 and $F_2$-IsoPs [33], and elevated levels of these predicted acute kidney injury (AKI) in obese patients in another study [35]. Similarly, IL-6 and $F_2$-IsoPs were elevated in exhaled breath condensate of obstructive sleep apnea (OSA) and obese patients [34]. Weight loss is associated with reduced BMI and anthropometric markers such as percent fat and fat mass, oxidative stress, $F_2$-IsoPs, and IL-6 [31]. Racial disparities exist in the plasma IL-6, with Blacks having higher levels than White populations [69]. Low socioeconomic status (SES), obesity, stressful life events, and depression are associated with elevated levels of IL-6 [70–72]. Thus, it is perhaps not surprising that several studies reported elevated IL-6 in Blacks compared to White populations [73–75]. Several studies showed that in addition to lymphocytes, adipocytes and adipose tissue monocytes secrete IL-6 in circulation [76,77]. High levels of IL-6 have been implicated in reducing insulin sensitivity and in an increased risk of T2D. Thus, our study suggests that high levels of IL-6 may contribute to reduced insulin sensitivity and oxidative stress in obese Black women. We summarized the findings of the studies in Figure 7.

Interestingly, our data show that obese women tend to consume less oxygen as measured by $VO_2$ (oxygen uptake) and MET (metabolite equivalent), and both parameters were inversely correlated with BMI, indicating hypoxic conditions in the deep and fat tissues (Figure 6E,F). Moreover, several studies suggested that adipogenesis, either by hyperplasia or hypertrophy, can cause hypoxia and a concomitant reduction in insulin sensitivity [78,79]. Hypoxia has also been implicated in inducing IL-6 production [33]. $F_ECO_2$ (fractional concentrations of expired carbon dioxide) and $F_EO_2$ (fractional concentrations of expired oxygen) exhibit positive and negative correlations, respectively, with $F_2$-IsoPs in obese Black women (Figure 6G,H). A higher concentration of $CO_2$ in expired air indicates

a higher rate of cellular respiration, while reduced oxygen in expired air indicates that the remaining $O_2$ might be used to generate free radicals such as superoxide ($O_2^-$). High levels of $F_2$-IsoPs in obese Black women are the 'footprints' of lipid peroxidation by reactive oxygen species such as superoxide.

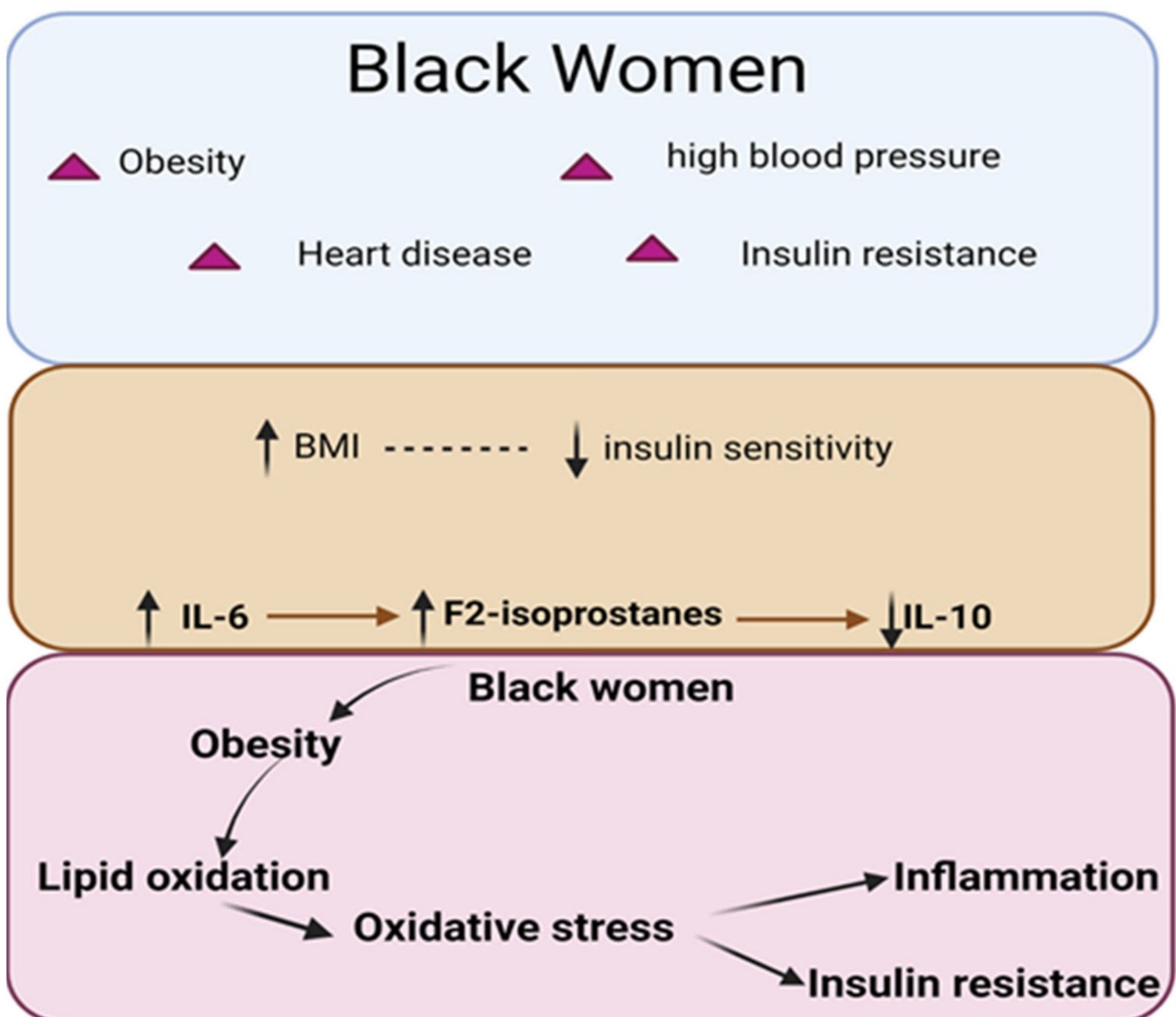

**Figure 7.** Graphical summary of the study. Eighty-eight obese Black women were recruited in this study. Pearson's correlation analysis revealed a positive correlation between obesity and reduced insulin sensitivity and increased F2-isoprostanes. F2-isoprostanes positively correlated with proinflammatory interleukin-6 (IL-6) while negatively correlated with interleukin-10 (IL-10).

A major strength of our study lies in the fact that it was designed for a specific group of obese Black women who is highly predisposed to obesity and insulin resistance, and T2D. Moreover, we employed state-of-the-art DXA to measure anthropometric and other parameters discussed in this study. However, the relatively lower number of participants constitutes a potential limitation of our study. Another potential limitation is the lack of comparative data in other ethnic groups, as well as the lack of data for low to normal BMI. In addition, our observational study and most of the studies in the literature are cross-sectional; the causality of the disproportional prevalence of obesity in Black women cannot be inferred.

Therefore, longitudinal and interventional studies are warranted for a better understanding of the mechanisms underlying the disproportionality of high prevalence of obesity, reduced insulin sensitivity, and T2D in obese Black women. It would be interesting to determine the involvement of adipose tissue in ROS production as one of the main culprits of reduced insulin sensitivity, increased insulin resistance, and increased risk of T2D in obesity. Moreover, there are differences in adipose tissue distribution in the body between Black and White women (gynoid vs android distribution, and SAT vs VAT); studying and understanding the contribution of each adipose tissue may reveal the individual contribution in oxidative stress, cytokines production, insulin sensitivity, and T2D.

**Supplementary Materials:** The following supporting information can be downloaded at: https://www.mdpi.com/article/10.3390/endocrines4010003/s1, Figure S1: Relationship between BMI and anthropometric DXA measurements. Our data show a robust positive correlation between BMI and the following variables: (A) DXA weight, (r = 0.92, $p$ = <0.0001), (B) VAT ($cm^3$, r = 0.48, $p$ = 0.0007), (C) e-VAT (g, r = 0.48, $p$ = 0.0007), (D) DXA fat mass (r = 0.66, $p$ = <0.0001), (E) DXA fat free mass (r = 0.66, $p$ = <0.0001), (F) DXA percent fat (r = 0.75, $p$ = <0.0001), (G) android percent fat (r = 0.63, $p$ = <0.0001), (H) gynoid percent fat (r = 0.43, $p$ = <0.003), (I) android total mass (kg, r = 0.56, $p$ = <0.0001), and (J) gynoid total mass (kg, r = 0.78, $p$ = <0.0001). Pearson's correlation analysis was employed to determine the correlation. $p$ < 0.05 was considered a statistically significant correlation. Figure S2: Relationship of various body parts and their fat content with BMI. There was a strong positive correlation between BMI and (A) arms percent fat (r = 0.69, $p$ = <0.0001), (B) legs percent fat (r = 0.53, $p$ = <0.0001), and (C) trunk (r = 0.68, $p$ = <0.0001). Similarly, there were strong positive correlations between BMI and (D) arms total mass (kg, r = 0.56, $p$ = <0.0001), (E) legs total mass (kg, r = 0.67, $p$ = <0.0001), and (F) trunk total mass (kg, r = 0.86, $p$ = <0.0001), (G) HOMA1-IR (r = 0.25, $p$ = <0.08), (H) HOMA2-IR (r = 0.14, $p$ = <0.32). Pearson's correlation analysis was employed to determine the correlation. $p$ < 0.05 was considered a statistically significant correlation. Figure S3: Relationship between BMI and certain body circumferences. Our data show strong positive correlations between BMI and circumferences of (A) waist (cm, r = 0.74, $p$ = <0.0001), (B) thigh (cm, r = 0.61, $p$ = <0.0001), and (C) hip (cm, r = 0.84, $p$ = <0.0001). Pearson's correlation analysis was employed to determine the correlation. $p$ < 0.05 was considered a statistically significant correlation. Figure S4: Relationship between anthropometric DXA measurements and F2-IsoPs. Our data show a positive correlation between (A) arms percent fat (r = 0.34, $p$ = 0.02), (B) legs percent fat (r = 0.37, $p$ = 0.01), (C) legs total mass (kg, r = 0.29, $p$ = 0.05), and (D) thigh circumferences (cm, r = 0.27, $p$ = 0.03) were correlated with F2-isoprostanes. Pearson's correlation analysis was employed to determine the correlation. $p$ < 0.05 was considered a statistically significant correlation.

**Author Contributions:** C.A.S. and A.K. conceived and designed the research; C.A.S. and G.L.M. performed experiments; M.S., C.A.S., P.M., P.D.K. and J.H.P. analyzed data; M.S., C.A.S. and A.K. interpreted results of experiments; M.S., P.D.K. and J.A.I. prepared figures; M.S. drafted the manuscript; M.S., A.K., C.A.S. and J.A.I. edited and revised manuscript; A.K. and C.A.S. approved the final version of the manuscript. All authors have read and agreed to the published version of the manuscript.

**Funding:** This work was supported by Dysautonomia International; the Vanderbilt Autonomic Dysfunction Center; Vanderbilt CTSA grant UL1TR002243 from NCATS/NIH, the American Heart Association grant to J.A.I. 903428 and the National Institutes of Health grants R03HL155041, and R01HL144941 to A.K.

**Institutional Review Board Statement:** The study was approved by an institutional review board (Vanderbilt Human Research Protection Program).

**Informed Consent Statement:** All participants gave written informed consent. Further, the study was conducted following institutional guidelines and adhered to the principles of the Declaration of Helsinki and Title 45 of the US Code of Federal Regulations (Part 46, Protection of Human Subjects).

**Data Availability Statement:** Not applicable.

**Acknowledgments:** The Vanderbilt University Eicosanoid Core Laboratory and GLM are supported by the Diabetes Research and Training Center (NIDDK Grant DK-20593).

**Conflicts of Interest:** No conflict of interest, financial or otherwise, are declared by the authors.

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
