# Peer review of "Obesity Is Associated with Increased F2-Isoprostanes and IL-6 in Black Women"

_endocrines, doi:10.3390/endocrines4010003_

Round 1

Reviewer 1 Report

1. This study resulting evidence supports F2-IsoPs may be associated with obesity-induced cardiovascular risk in black women by increasing production of inflammatory cytokine IL-6 and decreasing the production of anti-inflammatory IL-10. Theses results are interesting and study were well conducted.

2. Try to do more Discussion compared with the White Woman?

3. The format of references is not uniform. At the same time, more new literatures need to be cited.

 With minor revision as commented above, I recommend the publication in the Endocrine.

Author Response

Reviewer 1: Comments and Suggestions for Authors

1. This study's resulting evidence supports that F2-IsoPs may be associated with obesity-induced cardiovascular risk in black women by increasing the production of inflammatory cytokine IL-6 and decreasing the production of anti-inflammatory IL-10. These results are interesting, and the study was well conducted.

Answer: Thank you for this concise and accurate summary of our manuscript.

  1. Try to do more Discussion compared with the White Woman?

Answer: Thank you for your suggestions. We added the racial comparison studies in the ‘Discussion’ section. Please see the yellow-highlighted text in the manuscript with track changes.

  1. The format of references is not uniform. At the same time, more new literature needs to be cited.

Answer: The error has been amended, and new literature in the introduction and discussion has been incorporated. Thank you.

  1. With minor revision as commented above, I recommend the publication in the Endocrine.

Answer: Thank you.

Reviewer 2 Report

This manuscript tried to demonstrate that F2-IsoPs may be associated with obesity-induced cardiovascular risk in black women by increasing production of inflammatory cytokine IL-6 and decreasing the production of anti-inflammatory IL-10. The manuscript was designed and performed well. However, the following concerns may be required to improve this manuscript.

1. The study suggests  that F2-IsoPs may be associated with obesity-induced cardiovascular risk. The author should  introduce more relationship between F2-IsoPs and cardiovascular diseases and risk factors.

2. It looks that F2-IsoPs may be not specifically associated with obesity-induced cardiovascular risk. This results might be used to explain the effects of F2-IsoPs on many kinds of disease. So, more parameters about cardiovascular risk need to be measured.

3. More experiments in vitro will easy to strength the conclusions.

Author Response

Reviewer 2: Comments and Suggestions for Authors

This manuscript tried to demonstrate that F2-IsoPs may be associated with obesity-induced cardiovascular risk in black women by increasing the production of inflammatory cytokine IL-6 and decreasing the production of anti-inflammatory IL-10. The manuscript was designed and performed well. However, the following concerns may be required to improve this manuscript.

  1. The study suggests that F2-IsoPs may be associated with obesity-induced cardiovascular risk. The author should introduce more relationships between F2-IsoPs and cardiovascular diseases and risk factors. It looks that F2-IsoPs may not be specifically associated with obesity-induced cardiovascular risk. These results might be used to explain the effects of F2-IsoPs on many kinds of diseases. So, more parameters about cardiovascular risk need to be measured.

Answer: We have previously shown that hypertensive patients exhibit higher plasma F2-IsoP than normotensive individuals (Figure A, published in Kirabo et al. 2014 JCI). NT indicates normotensive, HT, well-controlled hypertensive, and RH, resistant hypertensive patients.

As suggested, we further analyzed if other cardiovascular risk factors correlate with plasma F2-IsoPs. Our further analysis showed that high-density lipoprotein (HDL) is inversely correlated with F2-IsoP in obese women with BMI between 30 and 40 (r= -0.3331, P<0.039). We also found a trend of a positive association between F2-IsoP and low-density lipoprotein (LDL) in obese women with a BMI between 30 and 40 (r= 0.1784, P<0.1386). These associations suggest that F2-IsoP is a cardiovascular risk factor.

  1. More experiments in vitro will be easy to strengthen the conclusions.

Thank you for this suggestion. In the revised manuscript, we have now included data where we measured blood pressure and F2-isoprostane in PBMCs isolated from obese black women. We found that Black women with normal systolic blood pressure equal or less than 120mmHg have lower F2 isoprostanes when compared to those with blood pressures above 120 mmHg (Shown below and in Fig 3E, p= 0.0496).

Reviewer 3 Report

This is a straightforward manuscript that investigated relationship among obesity, F2-IsoPs and IL-6 in obese black women. The manuscript needs to be carefully reviewed before submission. There are some wrong description or typing errors (page 4: line 162-163, 177, 187, 190). 

There are some other concerns:

1. In table 1, why some characteristics of volunteers were lacking (<88)?

2. The 'r' between IL-6 and BMI is 0.1441. In my opinion, the correlation is weak, though it reached statistically significant "0.0429".

3. In figure 6H, please check the result of pearson correlation analysis and the corresponding description in results and figure legend. 

Author Response

Reviewer 3: Comments and Suggestions for Authors

This is a straightforward manuscript that investigated the relationship among obesity, F2-IsoPs and IL-6 in obese black women. The manuscript needs to be carefully reviewed before submission. There are some wrong descriptions or typing errors (page 4: lines 162-163, 177, 187, 190).

Answer: Dear reviewer, we apologize for the errors. The errors have been amended.

There are some other concerns:
1. In table 1, why some characteristics of volunteers were lacking (<88)?

Answer: We did not measure all parameters in all participants as some did not show up for subsequent measurements. Nevertheless, the missing data from some participants do not affect the conclusion of the study.

  1. The 'r' between IL-6 and BMI is 0.1441. In my opinion, the correlation is weak, though it reached a statistically significant "0.0429".

Answer: Thank you and yes, we agree that the correlation is weak and although it reaches statistical significance, further studies are needed.

  1. In figure 6H, please check the result of the Pearson correlation analysis and the corresponding description in the results and figure legend.

Answer: Thank you; we have now corrected this in the revised manuscript.  

Round 2

Reviewer 2 Report

need to read carefully to correct minor errors.

Reviewer 3 Report

No more comment.